# Design and Synthesis of Free-Radical/Cationic Photosensitive Resin Applied for 3D Printer with Liquid Crystal Display (LCD) Irradiation

**DOI:** 10.3390/polym12061346

**Published:** 2020-06-15

**Authors:** Junyang Shan, Zijun Yang, Guoguang Chen, Yang Hu, Ying Luo, Xianming Dong, Wenxu Zheng, Wuyi Zhou

**Affiliations:** 1Research Center of Biomass 3D Printing Materials, College of Materials and Energy, South China Agricultural University, Guangzhou 510642, China; shanjunyang536@163.com (J.S.); yangzijune0606@163.com (Z.Y.); cgg@stu.scau.edu.cn (G.C.); huyang0303@scau.edu.cn (Y.H.); luoying@scau.edu.cn (Y.L.); dongxming@263.net (X.D.); 2Key Laboratory for Biobased Materials and Energy of Ministry of Education, College of Materials and Energy, South China Agricultural University, Guangzhou 510642, China

**Keywords:** 3D printing, photosensitive resin, liquid crystal display irradiation, UV-curing

## Abstract

In this work, aiming at a UV-curing 3D printing process with liquid crystal display (LCD) irradiation, a novel free-radical/cationic hybrid photosensitive resin was designed and prepared. After testing, the results showed that the acrylate monomers could be polymerized through a free-radical mechanism, while the epoxides were polymerized by a cationic curing mechanism. During the process of UV-curing, the acrylate and epoxide polymers were crosslinked and further locked together by non-covalent bonds. Therefore, an interpenetrating polymer network (IPN) structure could be formed through light-curing 3D-printing processes. Fourier transform infrared spectroscopy (FT-IR) revealed that the 3,4-epoxy cyclohexyl methyl-3,4-epoxy cyclohexyl formate and acrylic resin were both successfully involved in the UV-curing process. Furthermore, in order to make the 3D-printed objects cured completely, post-processing was of great importance. The results from the systematic study of the dynamic mechanical properties of the printed objects showed that the heating treatment process after UV irradiation was very necessary and favorable for the complete cationic polymerization of UV-6110 induced by Irgacure 261. The optimum heating treatment conditions were achieved at a temperature of 70 °C for 3 h.

## 1. Introduction

To date, as an additive manufacturing technology, 3D printing has been developed to build many 3D objects by a layer-by-layer printing process using some special materials, such as micron-sized powdery metals, thermoplastic or liquid photosensitive resins, and other adhesive materials [1,2,3,4,5]. Compared with the conventional subtractive manufacturing process, 3D printing has many advantages including a short manufacturing cycle, the capability of fabricating complex models [6], and reduced material waste. Thus, 3D printing has been applied to a wide range of fields, involving tissue engineering, the digitized manufacturing of metallic components, energy materials, and food processing. To date, 3D printing technologies have been developed rapidly and become a research hotspot in many countries all over the world [7,8,9,10,11]. Among all kinds of 3D printing technology, the stereolithography apparatus (SLA) rapid prototyping method is a typical 3D printing technology with a special control system using an ultraviolet (UV) laser beam to scan photosensitive polymers according to the profile data of each cross-section [10,11,12,13]. In a local region, the photosensitive resin absorbs the energy from UV light with special wavelengths and is then cured rapidly by a photo-polymerization process, and finally forms a thin layer [14]. The above process is repeated until the designed objects are completely formed.

Photosensitive resins are those materials that can be cured quickly under the excitation of UV light [15,16]. In general, the resins include reactive oligomers, reactive diluents such as reactive monomers, photoinitiators, filler pigments, and other auxiliaries. However, most of those compositions have strong odors and certain toxicities. In practical applications, the products need to be non-toxic and have no stimulative effect, with low odor. Therefore, there are few photosensitive resins that can be used for 3D printing. Furthermore, most of them are initiated by free radicals [17], which will lead to the hybrid resins being hard and brittle after curing, resulting in poor bending and stretching properties, severely limiting the range of application [18,19]. Used as the diluent of acrylic resin, N-acryloyl-morpholine (ACMO) is very stable with typically low irritative, corrosive, and toxic properties. As a key monomer, it can greatly enhance the mechanical performance of the cured resin. Furthermore, it has a rapid reaction rate and good flexibility. Polyurethane acrylate (PUA) is an oligomer with high wearability and good weather resistance properties [20,21,22]. During the UV light-curing process, the long-chain PUA can react with the ACMO to form a flexible spline owing to the low degree of cross-linking. After the 3D printing process, the conversion of reactive groups is usually incomplete, so the internal part of the sample is not cured completely. The curing rate of photosensitive resin is about 65–90 percent along with the incorporation of epoxy [23]. In order to complete the reaction, heat, ultrasonic energy, and UV light can be used in post processing [24,25]. The mechanical properties such as tensile strength and hardness can be effectively improved to a certain extent after post-curing treatment. 

At present, the thermal conductivity for the most of photosensitive resins is not uniform. In the post processing, the internal part of the products remains cool when the temperature is heated above its glass transition point, resulting in differential thermal strain and further cracking and warping [23].

In order to solve this issue, we explored the influence of the post-treatment (hot) conditions on the mechanical properties of 3D printing products formed by free-radical/cationic hybrid photosensitive resins.

## 2. Materials and Methods 

### 2.1. Materials

Acryl-morpholine (ACMO) was purchased from the Tianjin Jiuri Chemical Company (Tianjin, China). 3,4-epoxy cyclohexyl methyl-3,4-epoxy cyclohexyl formate (UV-6110) was purchased from the GuangZhou Kute Chemical Company (Guangzhou, China). Long-chain PUA and Antioxidant was donated by the Taiwan Changxing Chemical company (Taiwan, China). Phenyl bis(2,4,6-trimethylbenzoyl)-phosphine oxide (819) and (cumene) cyclopentadienyliron (ii) hexa-fluorophos (261) was obtained from the Changzhou Qiangli Chemical Company (Changzhou, China). Defoamer was donated by Byk (Wesel, Germany). 

### 2.2. Preparation of Photosensitive Resins and 3D Printing Tests 

After studying the tensile strength and elongation properties of the solidified resin samples by using varying ratios of PUA and ACMO, the optimal ratio of PUA to ACMO was confirmed as 3:7. The PUA/ACMO/UV-6110 hybrid resin was prepared by mixing different concentrations of PUA, ACMO, UV-6110, the free-radical photoinitiator Irgacure 819, and cationic photoinitiator Irgacure 261. The obtained photosensitive resins were stirred in a 3-neck flask at 50 °C for 30 min. A desk 3D printer with liquid crystal display (LCD) irradiation was used to print some designed objects for the evaluation of the mechanical properties of the resin. Three samples were designed and printed simultaneously with an inclination of 0° with respect to the printing platform. After printing, the formed samples were rinsed with ethanol in two successive baths for 15 min and then placed to dry. After that, the 3D-printed objects were post-cured in a dark vacuum oven at different temperatures. The heating treatment time and temperature for the post-curing process were set as 150 °C and 1 h, 100 °C and 1 h, 100 °C and 2 h, 80 °C and 1 h, 70 °C and 1 h, 70 °C and 2 h, and 70 °C and 3 h, respectively. 

### 2.3. Soxhlet Extraction Process

The gel content of the newly different 3D-printed samples, such as 100 °C 1 h, 70 °C 2 h, and 70 °C 3 h, was determined gravimetrically by five Soxhlet extraction cycles with the use of boiling THF (Tetrahydrofuran) as solvent (~5 h). Approximately, 0.1 g of 3D-printed sample was cut into small pieces and placed in a cellulose thimble. After that, the samples were dried in a vacuum oven at 50 °C till a constant weight was reached. The gel fraction was calculated using Equation (1), where W0 is the initial weight of the sample and W1 is the weight of the sample after extraction. For each sample, the gel fraction determination was repeated twice.
Gel Fraction (wt%) = (W1/W0) × 100(1)

### 2.4. Characterization of the Samples

The thermal properties of the liquid photosensitive resins and new 3D-printed samples treated as 100 °C for 1 h, 70 °C for 2 h, and 70 °C for 3 h, respectively, were measured with a MKIII Analyser of TA (New Castle, DE, USA). The loss and storage modulus and the loss tangent (tanδ, ratio of loss to storage modulus) were recorded as a function of the temperature. For these samples, the glass transition temperature was evaluated as the inflection point in the DSC curve observed. Of the samples, 5~15 mg of samples before or after post-curing process were put into different aluminum pans under a constant nitrogen flow of 50 mL min^−1^. All of the samples were heated from 0 °C to 130 °C with a heating rate of 5 k/min (frequency of 1 Hz) by using an extension mode. The thermal stability of those samples was investigated by thermogravimetric analysis (TGA). The decomposition temperature of the UV-cured test sample was measured with a TA-5000 Ir thermal analyzer at a heating rate of 10 °C/min. The heating temperature was set in a range from 25 to 600 °C with a heating rate of 10 °C·min^−1^. All the samples were heated and treated in nitrogen with a flow of 50 mL min^−1^ for non-oxidative degradation. The mechanical properties were characterized with a SHIMADZU universal testing machine. The dumbbell-shaped samples with a standard size of 150 mm × 10 mm × 4 mm were printed by a 3D printer with LCD light irradiation (LD-001, Shenzhen Chuangxiang 3D technology co. LTD). After that, the tensile strength of those samples was tested at a 5 mm min^−1^ stretching rate according to the national standard of GB 1040-2006. The rectangular samples with a size of 80 mm × 10 mm × 4 mm were printed for flexural tests with a deformation speed of 20 mm/min according to the national standard of GB 9341-2008. The fracture surface morphology of the printed objects was characterized by scanning electron microscopy (Hitachi S 4800). The acrylates part of the cured hybrid resins was selectively removed by using an alkaline solution of 10% KOH in methanol. The surface static contact angles of the samples were measured with an OCA 20 (Dataphysics, Stutt-gart, Germany) with the use of a 5 μL distilled water droplet placed automatically on the cured samples. The volume shrinkage of the samples was measured by reflective laser beam scanning. The Fourier transform infrared spectrophotometer (FT-IR) spectra of the UV-cured resins were recorded by using a Nicolet 5700 spectrometer with a range from 4000 to 650 cm^−1^ and a total of 64 scans at a resolution of 1 cm^−1^ in the transmission mode.

## 3. Results and Discussions

### 3.1. Mechanism of UV-Curing and LCD 3D Printing

In general, free-radical polymerization is made up of four continuous processes involving initiation, propagation, chain transfer, and termination. In the initiation process, the photo-initiator can transform from the ground state to the excited state after absorbing UV radiation, thus producing a chemical change to form some reactive intermediates of radicals, which can initiate the next polymerization between radicals and molecules of monomers or oligomers (e.g., with acrylates), forming reactive alkylmonomers. Those free-radical monomers with high reactivity can further react with other monomers or oligomers within the process of propagation. Chain transfer is a reaction process in which the reactive center interacts with a monomer or oligomer to produce another new free radical, which implies that the initial growth of the polymer radical has been terminated. The termination reaction is mainly completed by the interaction of two radicals, leading to the disappearance of the reactive center and the formation of stable polymers. One way of termination is the binding of two radicals in the active chain, and the other way happens through the disproportionation for bimolecular termination. Sometimes, the two ways happen simultaneously. The polymerization process ends when all the chemicals have reacted each other completely. The obvious fact is shown that the increase in the viscosity of the resins will hinder the movement of the molecular chains in spite of the presence of unreacted compounds [26]. In general, the cationic polymerization process also contains four steps. Differently from in the free-radical polymerization process, the cations act as the reactive center in the cationic polymerization. During UV light initiation, the cationic initiator can transform into an excited state, forming a new reactive center. For example, when using aryl-ferrocenium salts such as (cumene) cyclopentadienyliron(ii) hexa-fluorophos (Irgacure 261), the salt can be decomposed into cumene and ferrocene lewis acid under UV light irradiation. In terms of the hydrogen donor produced from the polyethers commonly used in the hybrid photosensitive resin, the aryl-ferrocenium cations produce protons combined with the anion and offer Brönsted acids, which can initiate the ring-opening reaction of epoxides. After reaction with Bronsted acids, the epoxide ring is opened and carbonium cations are produced, which can initiate the opening of other epoxide groups and lead to chain propagation [26]. The depicted mechanism of the cationic polymerization process is illustrated in Scheme 1.

The super acids produced by the photolysis of cationic photoinitiators can still initiate cationic polymerization after UV light irradiation or heat treatment. Therefore, the synthetic photosensitive resin with the use of Irgacure 261 can form an interpenetrating polymer network (IPN) structure through a post-curing process [27].

According to the imaging principle of a photosensitive 3D printer with LCD light irradiation, the image signals are provided by a computer program and displayed on a screen driving circuit [28]. Some transparent areas are selected and appear on the screen before irradiation. Under the irradiation with the LCD lights, the transparent region of the images displayed on the liquid crystal screen reduces the barriers to the LCD lights. However, in those regions without image display, the UV lights are blocked. The UV lights transmit through the LCD screen to form an area with images of UV lights. There is a liquid tank located on the surface of the liquid crystal screen. A transparent film is set at the bottom of the liquid tank. The LCD lights radiate directly to these liquid photosensitive resins through a transparent film so that those resins can be polymerized by the curing reaction and then solidified [29]. The opaque part of the LCD screen blocks the UV lights, and those resins without UV light irradiation still remain in the liquid state in the printing process. The display accuracy of the LCD screen is generally high. The resolution of the 3D printer that we used is 1280 × 768 pixels, and the dot matrix accuracy of the inch splay is 0.16 × 0.16 mm, which means that the size accuracy of the product formed through the LCD can reach 0.16 mm. The liquid crystal screen images from direct molding have a low deformation rate for graphics and high product accuracy. Typical cationic photopolymers, such as epoxides or oxetanes, are insensitive to oxygen but have the disadvantage of a low curing rate, resulting in some unreacted epoxies still existing in the printed objects. It was difficult to form a perfect IPN structure except by a post-curing process, such as heat treatment as shown in Scheme 2. 

### 3.2. FT-IR Analysis

Figure 1 shows the FTIR spectra of the cationic/radical initiated resin systems with different contents of epoxy under UV irradiation by a desk 3D printer with an LCD light source. For the uncured hybrid resins, the absorption features at 1634 cm^−1^ and 960 cm^−1^ correspond to the acrylates. The absorption feature at 914 cm^−1^ corresponds to the epoxy groups of 3,4-epoxy cyclohexyl methyl-3,4-epoxy cyclohexyl formate. It could be observed from the FTIR spectra that the absorption features of acrylates disappeared after 3D printing. The absorption peak at 1634 cm^−1^ disappeared in spectra b and e, which indicated that the double bonds reacted completely during the UV-curing of the 3D printing process. At the beginning of the UV irradiation, the absorption features of epoxy groups were also decreased to some extent. It could be concluded from the gradually decreasing intensity of the peak located at 914 cm^−1^ in spectra b to e that the UV-6110 had completed the ring-opening polymerization successfully. The strongest peaks located at 1720 cm^−1^ were attributed to the stretching vibration of the carbonyl groups in ester and carboxylic acid. In the present system, acrylate resins were polymerized through free-radical reactions while epoxides were polymerized by cationic mechanisms. Additionally, the comparison of the FTIR spectra between the cured hybrid photopolymers and the uncured acrylate or epoxy polymers showed that there was no new absorption peak appearing in the cured hybrid photopolymers, which meant that no reaction happened between the acrylate resin and epoxy resin. In terms of those UV light-cured hybrid resins, acrylate and epoxy polymers were locked together by cross-linking in a non-covalent manner. In the end, an IPN structure could be formed in the 3D-printed objects. As we know, the acrylate monomers were polymerized faster and more strongly than the epoxy monomers [30], which limited the diffusion ability of the epoxy molecules in the highly cross-linked polymeric network. A small part of the uncured epoxy resin did not participate in the UV-curing process. Therefore, in order to make the samples cured completely, a post-curing process was necessary for those 3D-printed objects.

### 3.3. Shrinkage of Photopolymer Resins

The contact angle of the 3D-printed samples to water was around 90° on the surface. As shown in Figure 2a, with an increasing content of epoxy resin, the wettability of the sample surface changed slightly, which meant that the contact angle did not clearly depend on the change in the UV-6110 concentration. It was clear that a medium-polarity surface was formed owing to the coexistence of hydrophobic materials (UV-6110) and the polar groups of acrylic oligomers, such as morpholinyl.

Under the UV light irradiation, liquid acrylate resins quickly transformed to a solid homogeneous film since the Van der Waals forces were converted into covalent bonds, significantly narrowing the average distance between the molecules. The epoxy resins could be incorporated into the acrylate resin system as fillers to decrease their shrinkage properties. It was found from Figure 2b and Table 1 that the shrinkage of the hybrid resins was obviously decreased along with the adding of epoxy.

### 3.4. Tensile Strength and Elongation Properties

The mechanical properties of the printed objects before and after post-curing were studied and analyzed, and the results are shown in Figure 3 and Table 2. The tensile strength and bending strength of the optimal samples before post-curing were 29 ± 1.4 MPa and 58± 2.1 MPa, respectively. The tensile strength and bending strength of the samples at 20% of epoxy monomers after post-cuing were 28.46 ± 0.5 MPa and 106 MPa, respectively. Therefore, the tensile strength of the samples after post-curing increased by up to 64% compared with the samples without post-curing. The bending strength increased up to 150%, which indicated that the mechanical properties of the 3D-printed samples were apparently improved after post-curing treatment.

When the epoxy resin content reached 20%, there were more open-ring bonds that reacted with the acrylate resin to form the IPN structure after the post-curing treatment, further increasing the tensile strength of the hybrid resins compared with that of the resins containing 10% of epoxy resins. However, when the epoxy resin content reached 30%, the efficiency of the cationic photo-initiator was insufficient under the post-curing at 100 °C for 2 h. The situation had reversed due to the facts that the IPN structure did not form completely in the samples and that those uncured epoxy resins acted as impurities, decreasing the tensile strength.

### 3.5. Effect of Heat Treatment at Different Temperatures on Spline Morphology

The heat treatment of 3D-printed objects was performed for 1 h at 150 °C, 100 °C, and 80 °C in a vacuum oven. The results shown in Figure 4 indicated that the printed samples were warped after heating, exhibiting higher shrinkage along the y-axis [31,32]. This observation revealed that excessive temperature during the post-curing process would induce parts of distortion. The reason was probably that while the external part of the sample heated up during the heat treatment, the internal part remained cool, leading to an uneven thermal change, which resulted in the external resins bearing bigger tensile stress. Ultimately, the differential thermal strains were generated, and cracking and warping were produced. As shown in Figure 5, many cracks were formed around the sample treated at 150 °C for 1 h.

In order to make 3D-printed products with high accuracy, the post-heating treatment should be carried out at an appropriate temperature. It was found that when the sample was treated at 70 °C for 3 h, no warp occurred, and the shrinkage ratios of the length and width were 0.97% and 1.5%, respectively, which indicated that a highly accurate sample was formed. 

### 3.6. Soxhlet Extraction

As a means of stereolithography 3D printing, UV light irradiation is very crucial and can initiate polymerization reactions or crosslinking between resins. To assess the degree of crosslinking, soxhlet extraction was employed to measure the gel content of the samples. The gel fraction (wt%) reflected the reaction degree of the resins after UV-curing and post-heating treatment. The experimental data are shown in Table 3 in detail. It could be observed that the gel content increased dramatically from 0 to 83.1 wt% after the 3Dprinting process, which meant that there were some uncured resins in the sample. After post-curing at 100 °C for 1 h, the gel content values of the samples increased up to 100 wt%, as they did after that at 70 °C for 3 h. The results revealed that the polymer chains of the resins had been reacted completely. Figure 6 shows that upon increasing the heating time from 1 h to 3 h at 70 °C, the tensile strength and bending strength were increased to 45 ± 1 MPa and 99 ± 2 MPa, respectively. Therefore, the hybrid resins did not react completely during the 3D printing process. It is necessary to make a perfect and stable product with the use of a heating treatment after 3D printing. In the current work, the hybrid resins made by us could prepare stable products by LCD light irradiation-based 3D printing with a post-curing treatment at 70 °C for 3 h. This was due to an integrated IPN structure being formed completely in the hybrid resins, resulting in excellent mechanical properties.

### 3.7. Thermal Stability of the 3D-Printed Objects

TGA analysis allows the studying of the thermal stability of the resins. The results of the TGA analysis are shown in Figure 7, and the degradation temperatures are listed in Table 4. Obviously, all of the samples after the post-heating treatment had a good heat resistance property. It was observed that the weight loss was less than 10.0 wt% as the temperature was increased up to 200 °C and then less than 20.0 wt% as the temperature was increased up to 300 °C. The weight loss in the first stage was attributed to those small molecules, such as photoinitiator 819 and 261, and moisture inside of the samples. The second stage happened in the range of 200–350 °C and could be regarded as the decomposition of the hard section where the post-cured sample reached a nearly constant weight. Therefore, in this stage, the unreacted acrylic resins and epoxy monomers were evaporated constantly while the internal part of the sample reacted completely. In the range of 350–450 °C, due to the decomposition of the cross-linked bonds and other chemical bonds, those uncured epoxy resins accelerated the thermal decomposition of the UV-cured resins as shown in Table 4. Although those resins had formed an IPN structure between the acrylate and epoxy resins in the process of UV-curing, the uncured epoxy monomers decreased their thermal stability. However, after post-curing, the hybrid resins initially increased the thermal stability due to the complete reaction of the hybrid resins and further formation of IPN structures between the acrylate resins and epoxy resins. 

### 3.8. Dynamic Mechanical Properties

Dynamic mechanical analysis (DMA) conducted in flexion mode was used to study the variation in the mechanical properties of the 3D-printed samples with changes in time and temperature. Moreover, DMA was also used to study the difference between the samples before and after post-curing. The thermal mechanical properties of the 3D-printed samples before and after post-curing were studied through a DMA test, and the results are shown in Figure 8 and Table 5, which contain the flexion storage modulus (E′) and tan δ measured for those fresh 3D-printed samples at 6 Hz frequencies.

As observed from the DMA curves of the hybrid polymer storage modulus loss factor tan δ, there was only one glass transition temperature (Tg) detected, suggesting only a single phase structure in the sample and no phase separation [33,34], which meant that the acrylic resin and epoxy resin had good compatibility under UV light irradiation. Epoxy molecular chains and the acrylic molecular chains interweaved with each other and prevented phase separation. Figure 8 showed that, compared with post-cured pure acrylic resins, the unreacted epoxy monomers decreased the thermal stability of the 3D-printed objects. After the post-curing process, the hybrid sample showed a higher glass transition temperature (Tg) than the pure acrylic resins, indicating that the thermal stability of the resins after post-heat treatment had been improved, probably due to the formation of IPN structures.

### 3.9. Finite Element Analysis of the Effect of the Temperature on the Spline Warp

In order to analyze the effect of the temperature on the spline warp, a finite element analysis (FEA) was carried out [35,36,37]. The FEA process was as follows: (1) Calculate the heat flux of the spline within 300 s under contact with the thermostatic heating baseplate; (2) Set the heat flux data as the load of the spline, and calculate the temperature field change within 300 s of the spline; (3) According to the data of the maximum temperature and the minimum temperature obtained in Step 2, select the load step with the maximum temperature difference to calculate the warping caused by the thermal expansion of the spline. The simulation model was a cuboid spline with a length of 80 mm, a width of 10 mm, and a height of 4 mm. A thin plate was added at the bottom of the model as the heating bottom plate of the dark oven. The average density of the material was set as 1.2 g·cm^−3^, which remained unchanged during the heat treatment. The Young’s modulus, Poisson’s ratio, and specific heat capacity of the material were 213.29 MPa, 0.35, and 1600 J·kg^−1^·K^−1^. The isotropic thermal conductivity of the material was 0.18 w·m^−1^·K^−1^, and the isotropic average thermal expansion coefficient was 4.8875·10^−5^ °C^−1^. The heating base plate defaults to structural steel. All the figures obtained from FEA calculations are illustrated in the Appendix A.

The heat treatment of the spline was to place the spline in a dark oven, and the spline was heated by the heating bottom plate without air convection and thermal radiation. Assuming that the spline was heated by direct contact with the bottom plate, the warping phenomenon of the spline could be better analyzed by calculating the heat flux of the spline.

Appendix A, show the heat flux maps of the spline when it just came into contact with the heating baseplate at 80 °C, 100 °C, and 150 °C. The greater the temperature difference between the heating baseplate and the spline, the greater the heat flux would be. At 80 °C, the maximum heat flux of the spline was about 7775 w·m^−2^, while at 150 °C, the maximum heat flux of the spline reached 17,160 w·m^−2^. The heating power of oven was generally large. When the initial temperature difference between the heating source and the spline was large, the heat flux of the spline would be relatively large. The spline belonged to the polymer material, the thermal conductivity was low, and under the high heat flux, the internal temperature difference easily occurred and caused warping.

When the heat flux was loaded onto the spline, the maximum internal temperature difference could be calculated, thus the maximum deformation could be calculated. By comparing Appendix A, it can be seen that under high heat flux, the higher the heating temperature was, the greater the temperature difference inside the spline would be. During heat treatment at 150 °C, the temperature difference inside the spline could be as high as 88 °C. Appendix A shows that the thermal deformation of the spline in the Y-axis direction could reach 1.5 mm at 80 °C, Appendix A shows that the thermal deformation of the spline in the Y-axis direction could reach 2.0 mm at 100 °C, and Appendix A shows that the thermal deformation of the bending spline in the Y-axis direction could reach 3.4 mm at 150 °C. By comparing Appendix A, it can be seen that the higher the heating bottom plate temperature was, the more serious the warping phenomenon would be. The deformation of the thermosetting materials was irreversible, and when the temperature difference of the spline became small gradually, the warping was reduced a little but did not disappear.

### 3.10. Microstructure of the Objects Printed by 3D Printer with LCD Light Source

Some typically complicated 3D objects were selected and fabricated by a 3D printer with an LCD light source. These samples were printed directly without any supporting structures. The printing accuracy of the objects was evaluated through calculating the sizes of the 3D models and actual printed objects. After that, those printed 3D objects were transferred to an oven for further thermal post-curing. The photographs in Figure 10a,b were the samples with a hollow structure and a honeycomb structure, respectively. Due to similar refractive index of 3,4-epoxy cyclohexyl methyl-3,4-epoxy cyclohexyl formate and acrylates, all of those printed objects looked transparent. Figure 9c was a printed black elk from blending with 0.05 wt% of carbon black powders in hybrid resins. After comparison of the dimensions between the printed objects and the 3D models, it was found that the printed objects owned excellent printing accuracy. The SEM images of the fracture surfaces of the printed samples are shown in Figure 10, where no particles were found on the fracture surface of the synthesized photosensitive hybrid resins, indicating that the compatibility of the acrylic resin with the epoxy resin was fine. As shown in Figure 10, the sample after post-curing treatment as shown in Figure 10b exhibited a smooth and glassy surface representing a homogeneous structure. However, the fracture surface of the sample without post-curing heat treatment showed rougher features and more micro-cracks than the post-cured samples, which led to a low mechanical strength.

## 4. Conclusions

In the present work, a kind of novel sensitive hybrid resin aimed at 3D printing with LCD light irradiation through a free radical/cationic dual curing process was prepared. The curing process involved an opening of the double bond of the acrylate resin and a ring-opening polymerization of the epoxy resin. Followed by further UV-curing, an interpenetrating polymer network (IPN) and a complex 3D shape were obtained. Due to the incomplete curing of the resins in the 3D printing process, the post-curing treatment was necessary and has been proved by this study in detail. The mechanical properties of the hybrid composite depended on the UV-curing degree. The optimal ratio of epoxy in the hybrid resin was 1:5, where the Tg value and storage modulus of the hybrid films increased after the heat treatment. The optimal post-curing conditions for the cationic polymerization were confirmed as a heating temperature of 70 °C and a heating time of 3 h. Both the thermal stability and mechanical strength of the synthesized free-radical/cational hybrid photosensitive resins were higher than those of the pure oligomer resins.

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
