# Peer review of "Design and Synthesis of Free-Radical/Cationic Photosensitive Resin Applied for 3D Printer with Liquid Crystal Display (LCD) Irradiation"

_polymers, 2020, doi:10.3390/polym12061346_

Round 1

Reviewer 1 Report

Dear Authors! I have reviewed your manuscript "Design and Synthesis of Free Radical/Cationic Photosensitive Resin Applied for 3D Printer with Liquid Crystal Display (LCD) Irradiation" and found that it can not be recommended for publishing in present form. The text is written and organized in inaccurate way so it is first of all uncomfortable for a reader due to modest English, many Typos and bad organizing of internal designation of samples, so it is very difficult to trace your logics. The LCD light and UV-light in the SLA 3D-printer are described in a very confusing manner, so a reader may understand that LCD irradiates UV waves. The speculations on the reasons of warpage are far from satisfactory and FEA is recommended to check your ideas on thermal gradient during post-curing heat treatment. Please, find other notes attached.   

Author Response

To Reviewer #1

(1) The text is written and organized in inaccurate way so it is first of all uncomfortable for a reader due to modest English, many Typos and bad organizing of internal designation of samples, so it is very difficult to trace your logics.

Response: Thank you very much for your kind reminding. The writing of the text and the organizing of internal designation of samples have been improved.

(2) The speculations on the reasons of warpage are far from satisfactory and FEA is recommended to check your ideas on thermal gradient during post-curing heat treatment. Please, find other notes attached.   .

Response: Thank you very much for your kind reminding. FEA has been used to detect thermal gradients during post-curing heat treatment. It is shown in section 3.9 (Finite element analysis on the effect of the temperature to the spline warp).

Reviewer 2 Report

Please see the attached PDF file

Author Response

Major points

(1) The hybrid polymer requires post-print thermal treatments for the full curing. There are plenty of existing 3D printed polymers that do not need post-print thermal curing. How do you make your approach attractive to the wider community in such a condition? Note that post-treatment is not a trivial task.

Response: Thank you very much for your kind reminding. Compared with those of pure acrylate system, the mechanical properties of the hybrid photosensitive resin are better. As shown in Figure 3, the mechanical properties of the hybrid resin are much better performing than that of the pure acrylic system.

(2)  The study clearly shows that the post-print thermal curing significantly enhances mechanical properties. Moreover, the 3D polymer has a massive curing-induced shrinkage (10-15% by volume). I am not sure how the approach will be lucrative to the wider audience.

Response: Thank you very much for your kind reminding. The purpose of this paper is to observe the effect of different post-curing temperature and time on the mechanical properties of hybrid photosensitive resin, so as to obtain the optimal post-curing conditions. So the shrinkage of resin is not the focus of this research. In other studies, resin shrinkage can be significantly reduced and printing accuracy can be improved by adding fillers, such as carbon black, nano silica, etc. Besides, Figure. 5 shows that the shrinkage of cured resin after post-curing for 3 hours at 70 ° C is 2%, which means that after post-curing, 3D-printed objects can still maintain certain accuracy.

(3) The authors mentioned as @ The volume shrinkage of the samples are measured by a reflective laser beam scanning@. I did not get the process. Please explain a bit more.

Response: Thank you very much for your kind reminding. Test principle of laser beam scanning is that,use a few ml or less of resin sample quantity to cure based on the light curing conditions, measure the thickness change of the sample before and after curing, and obtain the curing shrinkage rate of the resin through the reduction rate of the thickness.

(4) Please provide figures of tensile and flexural test set ups.

Response: Thank you very much for your kind reminding. The figures of tensile and flexural test set is SHIMADZU AG-X.

(5) What are the tolerance/reproducibility of each mechanical result? Put them in error bars or (plus/minus format)

Response: Thank you very much for your kind reminding. The error bars has been marked on the figures. Each data was retested by three times. The reason for the mechanical error is that the test splines are made of 3D printed, so the printed product will not exactly be the same as last time, so there will be errors in the mechanical property test.

(6) As authors mentioned curing is not finished during 3D printing with the LCD. How do measure the remaining reactive monomers that were not used/reacted?

Response: Thank you very much for your kind reminding. As is described in the section 2.3,through the Soxhlet extraction process, the remaining reactive monomers that were not reacted can be extracted out.

Gel Fraction (wt%) = (W1/W0) x 100      Eq. (1)

The gel fraction was calculated using Eq. (1), where W0 is the initial weight of sample and W1 is the weight of sample after extraction.

By calculation,the content of unreacted monomer can be obtained.

(7) Please note cure-induced shrinkage modelling and simulation is an active of current research. Hence, to widen the appeal of the current work, I would strongly recommend including some references from curing & shrinkage modelling community. a) Leistner et al.10.1016/j.polymertesting.2018.03.031, b) Hossain et al. doi.org/10.1007/s00466-008-0344-5,c) Wu et al. doi.org/10.1016/j.jmps.2017.11.018,d) Hossain et al. doi.org/10.1007/s00466-009-0397-0.

Response: Thank you very much for your kind reminding. Shrinkage modelling and simulation are shown in section 3.9, and those articles recommended by the reviewer have been quoted in the first sentence of section 3.9.

(8) Please mention the advantages of LCD light curing compared to other UV-curing techniques in the Introduction section.

Response: Thank you very much for your kind reminding. Advantages of LCD light curing: 1. High precision: it is easy to achieve a plane precision of 100 microns, which is superior to the first generation SLA technology and comparable with the current desktop level DLP technology. 2. Low price: mainly compared with SLA and DLP of previous generation technology, this cost performance is extremely outstanding. 3. Simple structure: because there is no Laser Galvanometer or projection module, the structure is simple and easy to assemble and repair.

Minor points

  1. Please correct Y axis of Figs 3, 6. It will be MPa (not Mpa)

Response: Thank you very much for your kind reminding. Error have been corrected. 

  1. What is the meaning of Red and Blue chains in Fig 2? Do you mean main monomers and crosslinkers?

Response: Thank you very much for your kind reminding. The blue segment is epoxy, while the red segment is acrylic.

  1. What is Da in Page 60?

Response: Thank you very much for your kind reminding. It's a mistake,The Da has been omitted.

Round 2

Reviewer 1 Report

Dear Authors! I am satisfied with your revised paper, and I think it has been improved. 

Reviewer 2 Report

It can be accepted now.